# Structure and engineering of *Brevibacillus laterosporus* Cas9
Toshihiro Nakane[1,10], Ryoya Nakagawa[1,10], Soh Ishiguro[2], Sae Okazaki[3], Hideto Mori[4,5,6], Yutaro Shuto [1], Keitaro Yamashita [3], Nozomu Yachie[2,6,7], Hiroshi Nishimasu [3,8,9] ✉ & Osamu Nureki [1] ✉

The RNA-guided DNA endonuclease Cas9 cleaves double-stranded DNA targets complementary to an RNA guide, and is widely used as a powerful genome-editing tool. Here, we report the crystal structure of *Brevibacillus laterosporus* Cas9 (BlCas9, also known as BlatCas9), in complex with a guide RNA and its target DNA at 2.4-Å resolution. The structure reveals that the BlCas9 guide RNA adopts an unexpected architecture containing a triple-helix, which is specifically recognized by BlCas9, and that BlCas9 recognizes a unique $N_4CNDN$ protospacer adjacent motif through base-specific interactions on both the target and non-target DNA strands. Based on the structure, we rationally engineered a BlCas9 variant that exhibits enhanced genome- and base-editing activities with an expanded target scope in human cells. This approach may further improve the performance of the enhanced BlCas9 variant to generate useful genome-editing tools that require only a single C PAM nucleotide and can be packaged into a single AAV vector for in vivo gene therapy.

CRISPR-Cas (clustered regularly interspaced short palindromic repeats and CRISPR-associated proteins) systems provide adaptive immunity against mobile genetic elements in bacteria and archaea[1]. Cas9 from *Streptococcus pyogenes* (SpCas9) associates with dual RNA guides (CRISPR RNA [crRNA] and *trans*-activating crRNA [tracrRNA] or their artificially connected single-guide RNA [sgRNA]) and cleaves double-stranded DNA (dsDNA) targets complementary to the RNA guide, using its HNH and RuvC nuclease domains[2,3]. Besides the guide RNA–target DNA complementarity, SpCas9 requires an NGG (where N is any nucleotide) protospacer adjacent motif (PAM), located downstream of the target sequence[3]. Since SpCas9 with its sgRNA can target endogenous genomic sites in a wide range of cell types and organisms, it has been widely used for numerous technologies, such as genome editing, transcriptional regulation, and epigenetic modulation[4]. Cas9 orthologs from different microbes function with their cognate guide RNAs, and recognize a variety of PAM sequences[5,6]. Thus, the use of Cas9 orthologs expands the target range in Cas9-mediated genome engineering.

Structural studies of various Cas9 orthologs, such as SpCas9[7,8], *Staphylococcus aureus* Cas9 (SaCas9)[9], *Francisella novicida* Cas9 (FnCas9)[10],

*Campylobacter jejuni* Cas9 (CjCas9)[11], *Corynebacterium diphtheriae Cas9* (CdCas9)[12], *Neisseria meningitidis* Cas9 (NmCas9)[13], and *Streptococcus thermophilus* Cas9 (St1Cas9)[14], have highlighted the mechanistic conservation of the CRISPR-Cas9 enzymes. Cas9 enzymes commonly adopt a bilobed architecture consisting of recognition (REC) and nuclease (NUC) lobes, with the guide RNA–target DNA heteroduplex accommodated within the positively charged central channel. The REC lobe mainly consists of α-helices and recognizes the RNA–DNA heteroduplex and the sgRNA scaffold, whereas the NUC lobe consists of the RuvC, HNH, Wedge (WED), and PAM-interacting (PI) domains. Cas9 enzymes recognize the PAM nucleotides by the PI domain, and cleave the target and non-target strands using the HNH and RuvC domains, respectively. Structural comparisons between the Cas9 orthologs also revealed the mechanistic diversity among the CRISPR-Cas9 enzymes[8–14]. Although they share similar domain organizations, their REC and WED domains are structurally divergent, thereby recognizing the distinct architectures of their cognate guide RNAs. In addition, their PI domains adopt a conserved core fold, but recognize different PAM sequences using specific sets of amino-acid residues.

[1]Department of Biological Sciences, Graduate School of Science, The University of Tokyo, 7-3-1 Hongo, Bunkyo-ku, Tokyo 113-0033, Japan. [2]School of Biomedical Engineering, Faculty of Applied Science and Faculty of Medicine, The University of British Columbia, Vancouver, BC V6S 0L4, Canada. [3]Structural Biology Division, Research Center for Advanced Science and Technology, The University of Tokyo, 4-6-1 Komaba, Meguro-ku, Tokyo 153-8904, Japan. [4]Institute for Advanced Biosciences, Keio University, Yamagata 997-0035, Japan. [5]Graduate School of Media and Governance, Keio University, Fujisawa, Kanagawa 252-0882, Japan. [6]Premium Research Institute for Human Metaverse Medicine (WPI-PRIMe), Osaka University, Suita, Osaka 565-0871, Japan. [7]Synthetic Biology Division, Research Center for Advanced Science and Technology, The University of Tokyo, Tokyo 153-8904, Japan. [8]Department of Chemistry and Biotechnology, Graduate School of Engineering, The University of Tokyo, 7-3-1 Hongo, Bunkyo-ku, Tokyo 113-8656, Japan. [9]Inamori Research Institute for Science, 620 Suiginya-cho, Shimogyo-ku, Kyoto 600-8411, Japan. [10]These authors contributed equally: Toshihiro Nakane, Ryoya Nakagawa. ✉e-mail: nisimasu@g.ecc.u-tokyo.ac.jp; nureki@bs.s.u-tokyo.ac.jp

*Brevibacillus laterosporus* Cas9 (BlCas9, also known as BlatCas9) reportedly recognizes a unique N$_4$CNDD (where D is A, T or G) or N$_4$CNAA as the PAM, and induces indels in maize[15] and mammalian cells[16]. Given that most Cas9 orthologs recognize G-rich sequences as the PAM, BlCas9 can target genomic sites inaccessible by other Cas9 orthologs. Moreover, BlCas9 consists of 1092 residues, and is 276-residues (~0.8 kb) smaller than SpCas9 (1368 residues). Thus, as compared with SpCas9, BlCas9 with its sgRNA can be more efficiently packaged into an adeno-associated virus (AAV) vector, making it a potentially valuable asset for in vivo therapeutic genome editing. However, the optimal guide length and PAM preference for BlCas9 have not been fully investigated in vitro. In addition, the PAM recognition mechanism of BlCas9 also remains elusive, due to the lack of structural information and the limited sequence similarity between BlCas9 and other structurally characterized Cas9 orthologs.

Here, we performed functional and structural characterizations of BlCas9. We confirmed that BlCas9 exhibits robust activity with an sgRNA with an optimal 22-nucleotide (nt) guide and recognizes N$_4$CNDN PAMs with a pronounced preference for A at positions 7 and 8. The crystal structure of the BlCas9–sgRNA–target DNA complex revealed the remarkable diversity in the sgRNA architecture and the PAM recognition mechanism. Furthermore, we successfully engineered a BlCas9 variant with enhanced cleavage activity and an expanded targeting scope by structure-based rational design.

## Results

### Biochemical characterization of BlCas9

While the Cas9 orthologs require different guide lengths for efficient DNA cleavage (20-, 21–23, and 22-nt guides are optimal for SpCas9, SaCas9, and CjCas9, respectively)[17,18], the optimal guide length for BlCas9 has not been fully characterized in vitro. To determine this parameter, we performed in vitro cleavage experiments using purified BlCas9, sgRNAs with 20–23 nt guide sequences (sgRNA20–23), which are complementary to three different targets (Targets 1–3), and their respective plasmid DNA targets with the 23 nt target sequence and a T$_3$CCCAA (Target 1) and T$_3$CCGAA (Targets 2 and 3) PAM (Fig. 1a). BlCas9 with all sgRNAs cleaved the three DNA targets, and sgRNA22 was superior for all three target sequences (Fig. 1a and Supplementary Fig. 1a). We next performed a PAM identification assay, using the purified BlCas9–sgRNA22 complex and a DNA library containing the target sequence (Target 1) adjacent to a randomized 8-bp sequence. The sequence logos of the 8 bp random sequences depleted in this assay showed that BlCas9 recognizes the N$_4$CNDD PAM, consistent with a previous report in which a PAM library was cleaved using a 20 nt guide sgRNA[15] (Fig. 1b). However, a detailed 2D profile focused on all 16 possible sequences at the 7th and 8th positions revealed that BlCas9 does not accommodate all combinations of DD at these positions, and requires an A at either one of them (Fig. 1c). To further examine the PAM preference of BlCas9, we measured the in vitro cleavage activities of the BlCas9–sgRNA22 complex toward target DNAs (Target 1) with 16 different PAMs, in which the fourth to eighth nucleotides in the canonical T$_3$CCCAA PAM were individually substituted (Fig. 1d and Supplementary Fig. 1b). BlCas9 efficiently cleaved the target plasmids with the T$_3$NCCAA and T$_3$CCNAA PAMs (Fig. 1d and Supplementary Fig. 1b), confirming that it has no preference for the 4th and 6th PAM nucleotides. In addition, it only cleaved the T$_3$CCCAA targets, but not the T$_3$CDCAA targets (Fig. 1d and Supplementary Fig. 1b), indicating the requirement of the 5th C for the PAM recognition. BlCas9 efficiently cleaved the target plasmids with the T$_3$CCCDA PAMs, but not the T$_3$CCCCA PAM, confirming the requirement of the 7th D for the PAM recognition (Fig. 1d and Supplementary Fig. 1b). BlCas9 cleaved the target plasmids with T$_3$CCCAN PAMs, but showed a preference of A > T = G > C at the 8th position (Fig. 1d and Supplementary Fig. 1b). Together, these results indicated that BlCas9 recognizes N$_4$CNDN as the PAM, and prefers A at both the 7th and 8th positions.

### Crystal structure of the BlCas9–sgRNA–DNA complex

To elucidate the PAM recognition mechanism of BlCas9, we attempted to determine the crystal structure of BlCas9 (1092 residues) in complex with an sgRNA and its target DNA, but failed to obtain crystals. Previous studies revealed that the HNH domain of Cas9 is mobile and dispensable for DNA recognition[7,19,20], suggesting that the HNH domain may hamper crystallization. We thus crystallized a BlCas9-ΔHNH variant, in which the HNH domain (residues 504–669) is replaced by a GGGSGG linker, as in the case of CjCas9[11] (Fig. 2a). After extensive crystallization screening, we determined the crystal structure of BlCas9-ΔHNH in complex with a 110-nt sgRNA, a 28-nt target DNA strand, and an 8-nt non-target DNA strand with the T$_3$CCAAA PAM, at 2.4-Å resolution (Fig. 2a–c, and Table 1).

The crystal structure revealed that BlCas9 adopts a bilobed architecture consisting of the α-helical REC lobe and the NUC lobe, with the sgRNA–target DNA heteroduplex bound within the central channel between the two lobes, as in the other Cas9 structures[7–14] (Fig. 2c and Supplementary Fig. 2). The REC lobe comprises the REC1 (residues 77–229) and REC2 (residues 230–453) domains, whereas the NUC lobe includes the RuvC (1–44, 454–503, and 670–806), WED (residues 821–920), and PI (residues 921–1092) domains. The RuvC domain consists of three separate motifs (RuvC-I–III), with RuvC-I and RuvC-III connected to the REC1 and WED domains via an arginine-rich bridge helix (residues 45–76) and a phosphate-lock loop (residues 807–820), respectively, as in the other Cas9 orthologs[7–14]. Consistent with the classification of both BlCas9 and CjCas9 in the type II-C category, the overall structure of BlCas9 is more similar to that of CjCas9[11] (PDB: 5X2G, root-mean-square deviation [RMSD] of 2.1 Å for 622 equivalent Cα atoms) than those of other Cas9 orthologs, such as SpCas9[8] (PDB: 4UN3, RMSD of 3.1 Å for 518 equivalent Cα atoms) and SaCas9[9] (PDB: 5CZZ, RMSD of 3.1 Å for 638 equivalent Cα atoms) (Supplementary Fig. 2).

The sgRNA guide segment (G1–C20) and the target DNA strand (dG1–dC20) form the RNA–DNA heteroduplex, which is bound within the positively charged central channel between the REC and NUC lobes (Fig. 2c–e). The target DNA strand (dA(−1)–dT(−8)) and the non-target DNA strand (dT1*–dA8*) form the PAM duplex, which is bound between the WED and PI domains (Fig. 2c–e). As in the other Cas9 structures, the phosphate backbone of the sgRNA seed region (C13–C20) is extensively recognized by the bridge helix and the REC1 domain, while the backbone phosphate group between dG1 and dA(−1) in the target DNA strand is recognized by the phosphate-lock loop (Fig. 2c–d). These conserved structural features indicate that the RNA-guided DNA cleavage mechanism of BlCas9 is similar to those of the other Cas9 orthologs.

### Structure and recognition of the sgRNA scaffold

The sgRNA comprises the guide segment (G1–C20), the repeat:antirepeat duplex (G21•U50–U33:A38), the tetraloop (G34–A37), and the tracrRNA scaffold (A51–U110) (Fig. 3a, b). A57–A60 are disordered, probably due to their flexibilities. Notably, the present structure revealed that the BlCas9 tracrRNA scaffold contains a triple-helix structure within two stem loops (stem loops 1 and 2) and two stems (stems 1 and 2), which was neither predicted from its primary sequence nor observed in the other Cas9 orthologs (Fig. 3a, b and Supplementary Figs. 3 and 4). As expected from the nucleotide sequence, the repeat:antirepeat duplex adopts the A-form-like conformation, which consists of four non-canonical base pairs (G21•U50 and U28•G43–C30•U41) and nine Watson-Crick base pairs (C22:G49–C27:C44 and C31:G40–U33:A38), and is recognized by the bridge helix and the REC1/WED domains (Fig. 3a–c). In particular, C30•U41 forms hydrogen bonds with Lys886, indicating the importance of C30•U41 for base-specific repeat:antirepeat recognition by BlCas9 (Fig. 3d). Stem loop 1 (A52–G66) is formed via four Watson-Crick base pairs (G53:C65–C56:G61) and a non-canonical base pair (A52•G66), and is recognized by the REC1 domain and the bridge helix (Fig. 3a–c). A62 is flipped out from the stem loop and forms hydrogen bonds and stacking interactions with Arg69 and Arg227, respectively (Fig. 3e). Stem 1 (A67–G71 and C80–U84) and stem 2 (G74–A78 and U104–C108) form a triple-helix structure, which is stabilized by two base triples, G71:C80•C103 and U72•A78:U104 (Fig. 3a–c and Supplementary Fig. 3a, b). U73 hydrogen bonds with the backbone phosphate of G76, while U79 hydrogen bonds with the main chain of

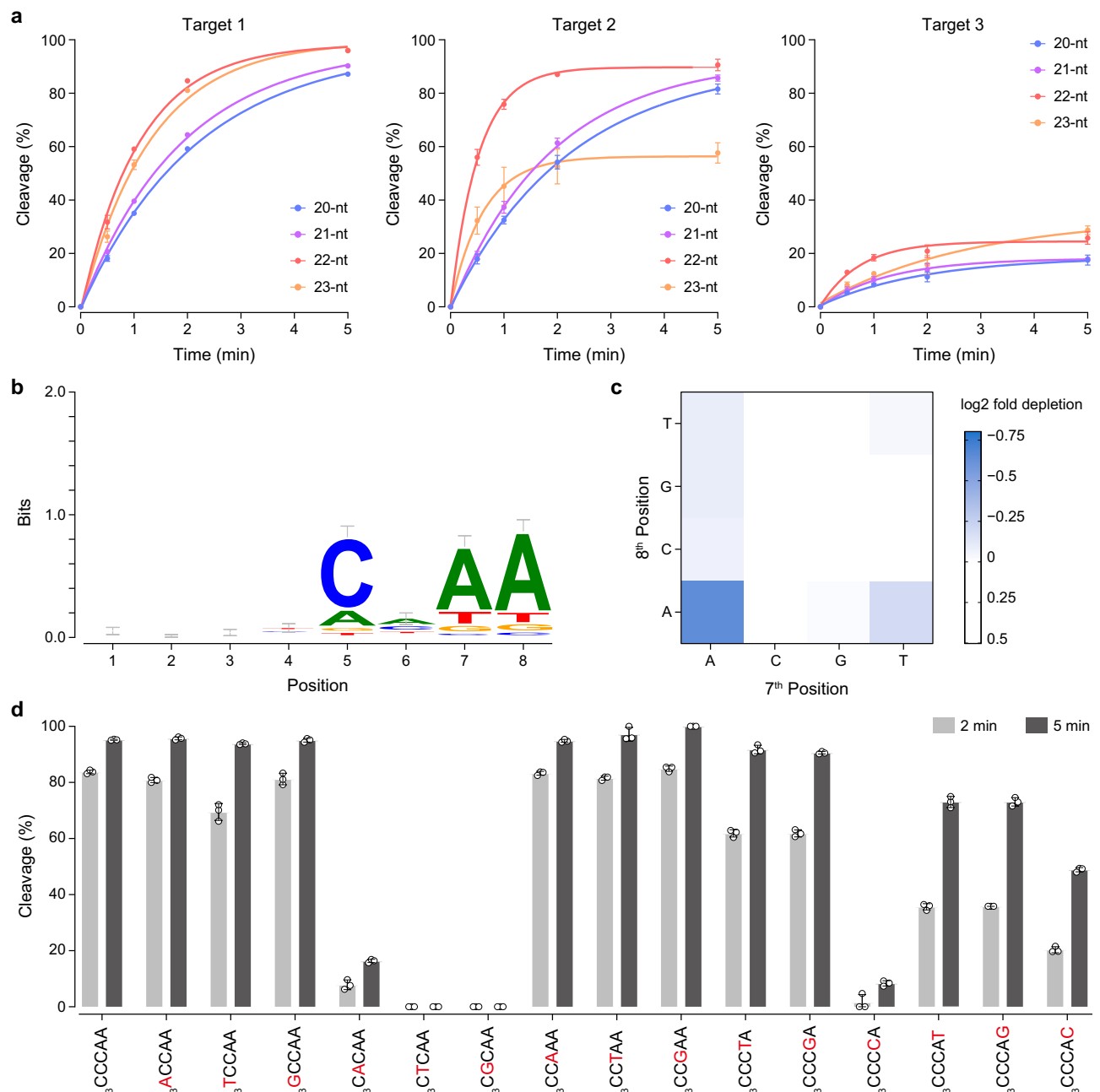

**Fig. 1 | In vitro cleavage activity. a** In vitro DNA cleavage activities of BlCas9 with the 20–23 nt guide sgRNAs toward three different target sequences (Targets 1–3). The linearized plasmid target bearing the $T_3CCCAA$ (Target 1) or $T_3CCGAA$ (Targets 2 and 3) PAM was incubated with the BlCas9–sgRNA complex at 37 °C for 0.5, 1, 2, and 5 min. The cleavage products were then analyzed by a MultiNA microchip electrophoresis system. Data are mean ± s.d. ($n$ = 3). **b, c** Sequence logo (**b**) and 2D profile (**c**) of the BlCas9 PAM obtained from the PAM identification assay. **d** In vitro DNA cleavage activities of BlCas9 with the 22-nt guide sgRNA toward DNA targets (Target 1) with different PAMs. The linearized plasmid targets were incubated with the BlCas9–sgRNA complex at 37 °C for 2 and 5 min. Data are mean ± s.d. ($n$ = 3).

Glu1071/Glu1073 in the PI domain (Fig. 3f). Stem loop 2 (A85–U102) is formed via seven Watson-Crick base pairs (A85:U102–U91:A96), and is recognized by the REC2 and RuvC domains (Fig. 3a–c). In particular, U95 is flipped out of the stem and forms hydrogen bonds with Glu248 and Lys427 in the REC2 domain (Fig. 3g). Taken together, the BlCas9 sgRNA adopts a unique conformation for recognition by BlCas9.

## PAM recognition

In the present structure, the PAM duplex is bound between the WED and PI domains (Fig. 4a). The nucleobases of dT1*–dC4* and dA6* do not directly contact the protein, consistent with the lack of specificity for positions 1–4

and 6 in the $N_4CNDN$ PAM. Importantly, the N4 of dC5* and the N7 and O6 of dG(−5) form hydrogen bonds with Asp1022 and Lys1040, respectively (Fig. 4b, c), explaining the observed requirement for the 5th C in the $N_4CNDN$ PAM. While the nucleobase of dA7* in the non-target strand is not recognized by the protein, the methyl group of dT(−7) in the target strand forms van der Waals interactions with Thr1025 and Ala1027 (Fig. 4b, d), explaining the preference for A at position 7. Similarly, the dA8* nucleobase in the non-target strand does not contact the protein, whereas the O4 of dT(−8) forms a hydrogen bond with Lys959, consistent with the observed preference for A at position 8 (Fig. 4b, d). The single mutations of Asp1022, Lys1040, and Lys959 abolished or reduced the in vitro DNA

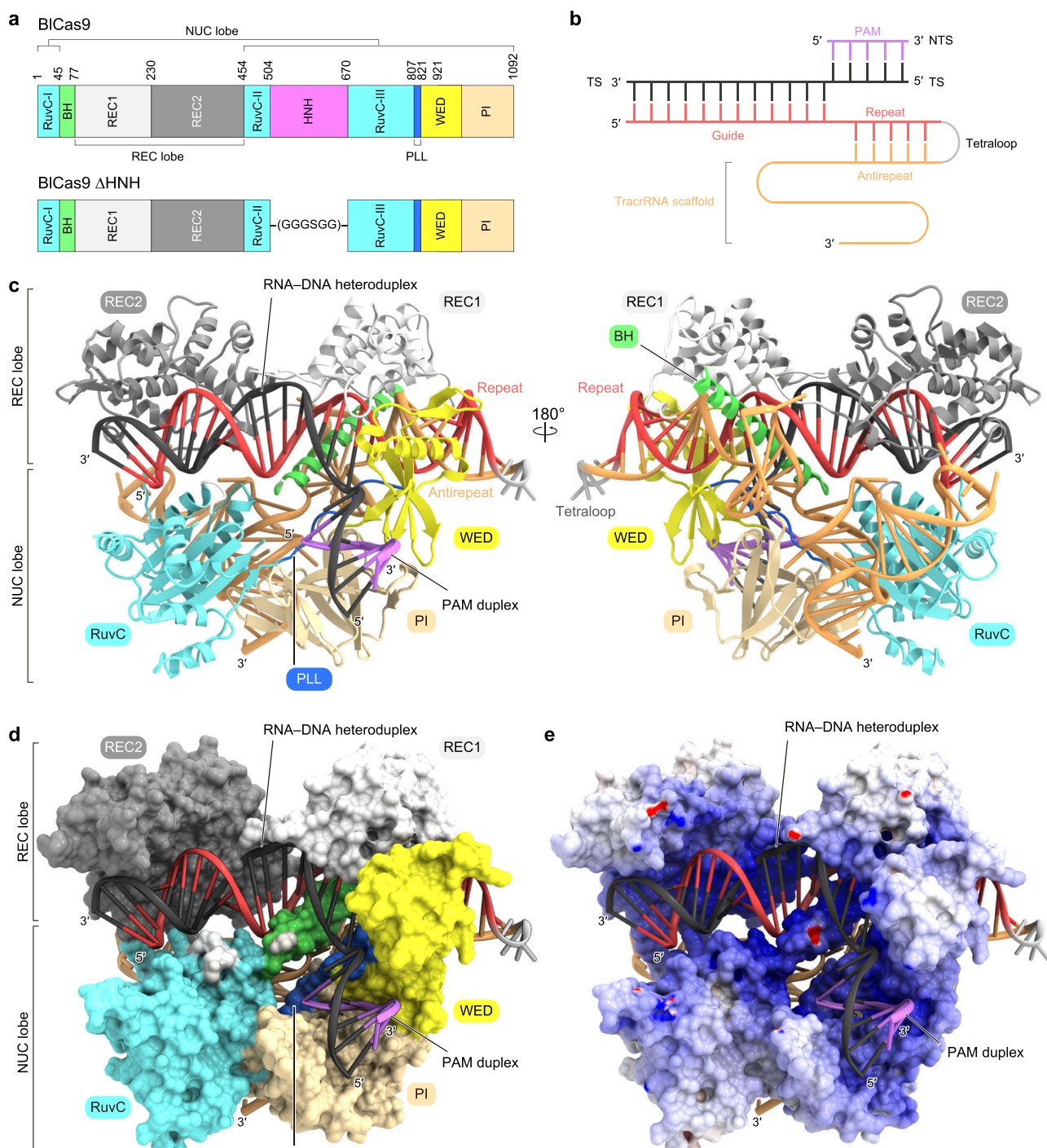

**Fig. 2 | Overall structure of the BlCas9–sgRNA–target DNA complex. a** Domain structure of BlCas9. The HNH nuclease domain was truncated for crystallization. BH bridge helix, PLL phosphate lock loop. **b** Diagram of the sgRNA and target DNA used for crystallization. TS target strand, NTS non-target strand. **c** Overall structure of BlCas9-ΔHNH in complex with the sgRNA and its target DNA. Disordered regions are indicated by dotted lines. **d**, **e** Surface representations of the BlCas9–sgRNA–target DNA complex, colored according to the protein domain (**d**) and electrostatic surface potential (**e**).

cleavage activities (Fig. 4e and Supplementary Fig. 5a), confirming the functional importance of these residues for PAM recognition. By contrast, the T1025A mutation did not reduce the cleavage activity, but rather relaxed the PAM preference at the 7th and 8th positions (Fig. 4e, f and Supplementary Fig. 5a,b), suggesting that the interaction between Thr1025 and dT(−7) is not crucial for PAM recognition. Taken together, these structural and functional analyses revealed that BlCas9 recognizes long, promiscuous PAM sequences through base-specific interactions with both the target and non-target strands. However, the present structure could not fully explain

the tolerance of T/G and the intolerance of C at the 7th position in the $N_4CNDN$ PAM. Therefore, additional structures with different PAM sequences are required to fully elucidate the PAM recognition by BlCas9.

## Molecular engineering

To expand the target range of BlCas9, we sought to engineer a BlCas9 variant with relaxed PAM preference at the 7th and 8th positions. Previous studies demonstrated that additional interactions between Cas9 and nucleic acids augmented the DNA cleavage activity[10,18,21]. Molecular modeling suggested

**Table. 1 | Crystallographic data collection, model refinement and validation**

| Data collection and processing | |
|---|---|
| Sample | BlCas9–sgRNA–target DNA |
| PDB ID | 8X5V |
| Beamline | SPring-8 BL41XU/SLS X06SA |
| Wavelength (Å) | 1.0 |
| Space group | *C2* |
| Cell dimensions | |
| a, b, c (Å) | 144.8, 99.2, 119.3 |
| β (°) | 97.1 |
| Resolution (Å) | 50–2.0 (2.12–2.0) |
| $R_{meas}$* | 0.255 (4.186) |
| *I/σI* | 19.01 (1.33) |
| CC(1/2)* | 0.999 (0.649) |
| Completeness (%)* | 99.9 (99.8) |
| Multiplicity* | 28.0 (28.2) |
| **Refinement** | |
| No. reflections | 107,265 |
| $R_{work}$/$R_{free}$ | 0.1948/0.2333 |
| No. atoms | |
| Protein | 7172 |
| Nucleic acid | 2983 |
| Others | 451 |
| *B*-factors (Å²) | |
| Protein | 59.2 |
| Nucleic acid | 63.1 |
| Others | 59.9 |
| R.m.s. deviations | |
| Bond lengths (Å) | 0.008 |
| Bond angles (°) | 1.787 |
| Ramachandran plot | |
| Favored (%) | 97.28 |
| Allowed (%) | 2.72 |
| Outliers (%) | 0.00 |

*Friedel pairs are treated as different reflections.

that Arg904 (E904R) forms a new interaction with the backbone phosphate of dA(−1) in the target strand (Supplementary Fig. 6a). Indeed, the E904R mutation enhanced the DNA cleavage activity of BlCas9 (Supplementary Fig. 6b). Thus, we measured the in vitro cleavage activities of the E904R/T1025A variant towards target plasmids with the $T_3CCCNA$ and $T_3CCCAN$ PAMs. The E904R/T1025A variant efficiently cleaved all of the $T_3CCCNN$ targets, including $T_3CCCCA$, for which the wild-type BlCas9 (referred to as BlCas9 for simplicity) exhibits almost no activity (Fig. 4f and Supplementary Fig. 5b). We hereafter refer to the E904R/T1025A variant as the enhanced BlCas9 (enBlCas9). To comprehensively analyze the PAM specificity of enBlcas9, we performed the PAM identification assay. In comparison to BlCas9, enBlCas9 showed some preference for the 8th position, but exhibited more relaxed PAM recognition at the 7th and 8th positions (Supplementary Fig. 6c, d). Together, these results demonstrated that our engineered enBlCas9 improves the cleavage activity and expands the target range as compared to BlCas9.

## BlCas9-mediated genome and base editing in human cells
To assess the activities of BlCas9 and enBlCas9 in mammalian cells, we measured indel formation induced by BlCas9 and enBlCas9 at 21

endogenous target sites with $N_4CNAN/N_4CNNA$ PAMs in human embryonic kidney (HEK) 293Ta cells. BlCas9 induced indels at 8 out of 21 target sites (at >1% frequencies) with an average frequency of 6.9%, whereas enBlCas9 induced indels at 12 out of 21 target sites with an average frequency of 10.8% (Fig. 5a). These results demonstrated that, consistent with our in vitro data, enBlCas9 exhibits higher cleavage activities than BlCas9 at several target sites in HEK293T cells.

Finally, we investigated the applicability of BlCas9 to base editing techniques in mammalian cells. Target-AID, comprising the SpCas9 D10A nickase mutant fused to the *Petromyzon marinus* cytosine deaminase 1 and uracil DNA glycosylase inhibitor, mediates C-to-T conversion at target genomic sites[22]. We replaced the SpCas9 D10A nickase in Target-AID with the D8A nickase version of BlCas9 or enBlCas9 to create BlCas9-AID and enBlCas9-AID, respectively, and then measured C-to-T conversions at 21 target sites (identical to those tested for indel formation) in HEK293Ta cells. BlCas9-AID induced C-to-T conversions at 7 target sites at >1% frequencies with an average frequency of 7.3%, whereas enBlCas9-AID induced them at 12 target sites at >1% frequencies with an average frequency of 8.5% (Fig. 5b). These results indicated that both BlCas9 and enBlCas9 can be utilized for base editing technologies, with enBlCas9 being more advantageous.

## Discussion
In this study, we determined the crystal structure of the BlCas9–sgRNA–target DNA complex, providing high-resolution insights into its sgRNA architecture and PAM recognition. The BlCas9 sgRNA contains the conserved repeat:antirepeat duplex, while the tracrRNA scaffold adopts an unpredicted triple-helix structure, which is not observed in the other Cas9 orthologs. Although the CjCas9 tracrRNA scaffold also possesses a triple-helix structure, their sequences and architectures are substantially different[11] (Supplementary Fig. 4). In addition, the triple-helix structure of CjCas9 is recognized mainly by the bridge helix, whereas that of BlCas9 is recognized by the RuvC and PI domains (Supplementary Figs. 2 and 4). These structural differences enable the species-specific recognition of their cognate tracrRNA scaffolds. The present structure also revealed the unique PAM recognition mechanism by BlCas9. Notably, BlCas9 forms hydrogen bonds with the C:G base pair at position 5 in the $N_4CNDN$ PAM, thereby identifying the characteristic C in the PAM. While the diverse PAM recognition mechanisms of the Cas9 orthologs have been reported, the mechanism primarily relying on single base-pair recognition is unique to BlCas9, thereby highlighting the diversity of Cas9-mediated PAM recognition.

We found that BlCas9 and BlCas9-AID with optimal 22 nt guide sgRNAs can induce indel formation and C-to-T conversion in human cells, suggesting their utility as in vivo genome editing tools. BlCas9 displayed significant variations in indel and C-to-T conversion efficiencies among different target sites with identical PAMs (Fig. 5a, b), indicating that the genome editing efficiencies are substantially affected by the genomic context, as observed previously[15]. Based on the structural information, we created the enBlCas9 (E904R/T1025A) variant with improved activity and slightly expanded targeting range. While several Cas9 and Cas12 orthologs have been reported to exhibit genome-editing activities in mammalian cells, most Cas9 and Cas12 orthologs require G- and T-rich sequences as their PAMs, respectively, thereby restricting their targetable genomic loci. In contrast, enBlCas9 can induce genome- and base-editing at target sites without G or T, potentially enabling applications in the treatment of genetic diseases that were previously inaccessible. In addition, since enBlCas9 (1092 residues) is much smaller than SpCas9 (1368 residues), enBlCas9 fused to a compact adenine/cytosine deaminase could be packaged into a single AAV vector for in vivo therapeutic base-editing[23,24]. Furthermore, we recently developed an approach that combines structure-informed design and deep mutational scanning to engineer variants with enhanced activity in a more reliable and efficient manner[22,25]. This approach may further boost the performance of the enBlCas9 variant to generate useful genome-editing tools that require only a single C PAM nucleotide and can be packaged into a

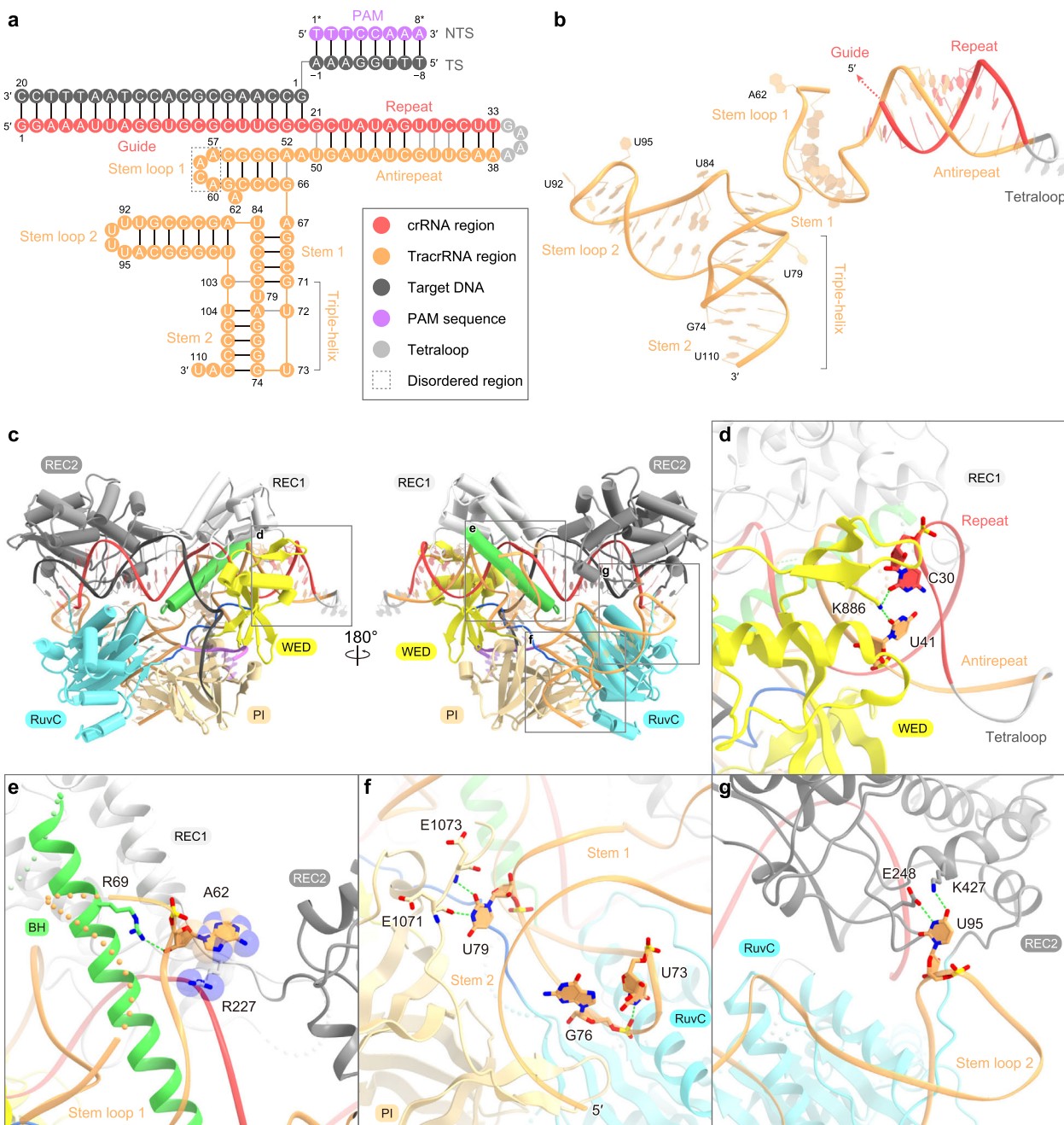

**Fig. 3 | Guide RNA architecture and recognition. a** Schematic of the sgRNA and target DNA. Disordered regions are enclosed in gray boxes. **b** Structure of the sgRNA scaffold. The disordered regions are indicated by dotted lines. **c** Recognition of the sgRNA scaffold by BlCas9. **d–g** Recognition of the repeat:antirepeat duplex (**d**), stem loop 1 (**e**), stems 1 and 2 (**f**), and stem loop 2 (**g**) of the sgRNA scaffold. Hydrogen bonds are depicted with green dashed lines.

single AAV vector. Collectively, our data highlight the structural and mechanistic diversity among the type II CRISPR-Cas9 effector enzymes, and pave the way for the development of a more compact genome editing toolbox.

## Methods

### Sample preparation

The gene encoding full-length BlCas9 (residues 1–1092) was codon optimized, synthesized (Genscript), and cloned between the *Nde*I and *Xho*I sites of the modified pE-SUMO vector (LifeSensors). The mutations were introduced by a PCR-based method, using the vector encoding full-length BlCas9 as the template, and the sequences were confirmed by DNA sequencing. For in vitro cleavage experiments, the N-terminally His$_6$-tagged

BlCas9 proteins were expressed in *Escherichia coli* Rosetta2 (DE3) (Novagen). The BlCas9-expressing *E. coli* Rosetta2 (DE3) cells were cultured at 37 °C in LB medium (containing 20 mg/l kanamycin) until the OD$_{600}$ reached 0.8, and protein expression was then induced by the addition of 0.1 mM isopropyl-ß-D-thiogalactopyranoside (Nacalai Tesque). The *E. coli* cells were further cultured at 20 °C for 18 hr, and harvested by centrifugation at 5000 g for 10 min. The *E. coli* cells were resuspended in buffer A (50 mM Tris-HCl, pH 8.0, 20 mM imidazole, and 500 mM NaCl), lysed by sonication, and then centrifuged at 10,000 g for 10 min. The supernatant was mixed with 0.3 ml Ni-NTA Superflow resin (QIAGEN) equilibrated with buffer A, and the mixture was loaded into a Poly-Prep Column (Bio-Rad). The protein was eluted with buffer B (50 mM Tris-HCl, pH 8.0, 300 mM imidazole, and 500 mM NaCl), and then the concentration of NaCl was

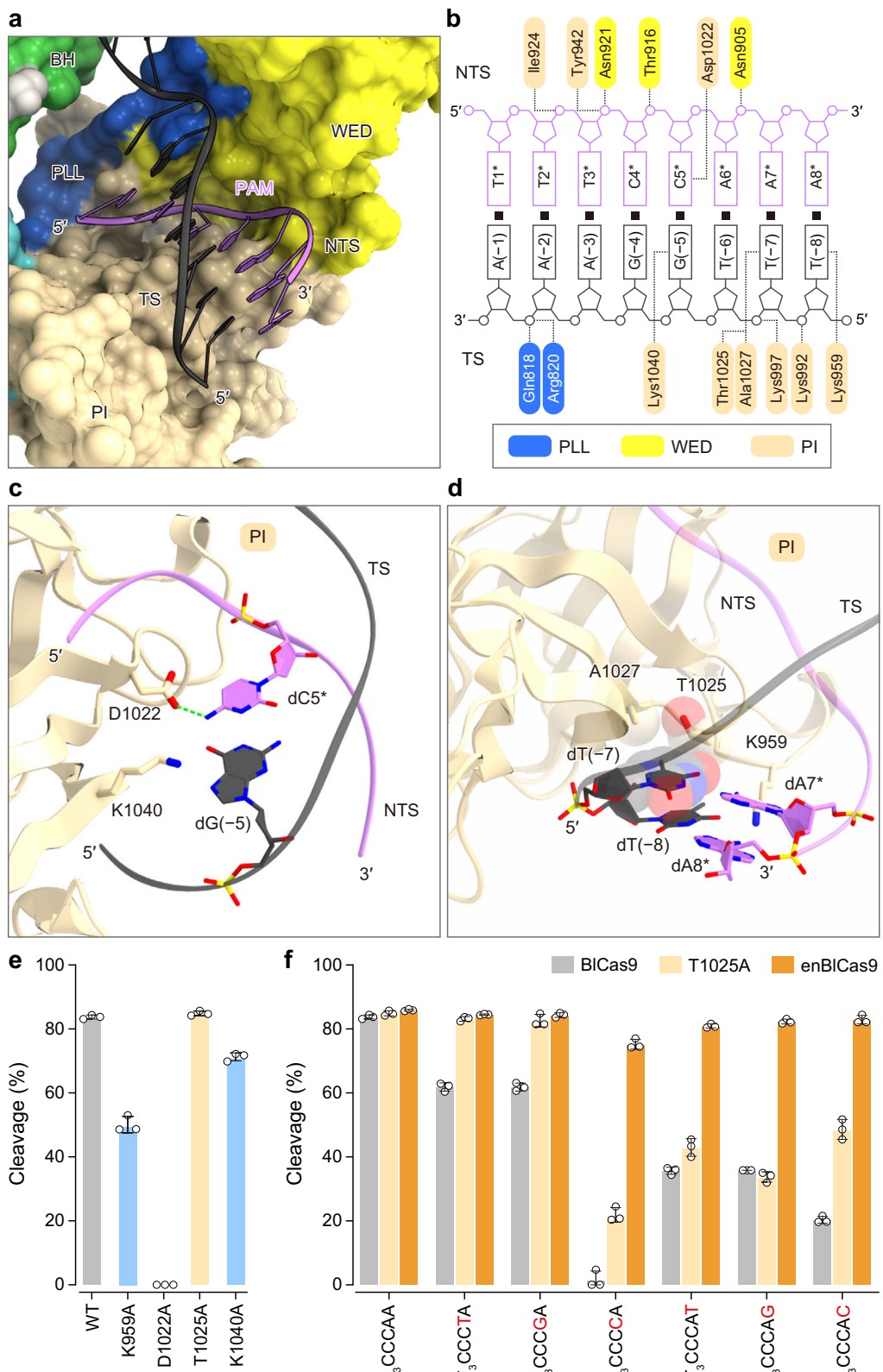

**Fig. 4 | PAM recognition. a** Binding of the PAM duplex to BlCas9. **b** Schematics of the PAM recognition by BlCas9. Hydrogen bonds are depicted by dashed lines. **c** and **d** Recognition of the $N_4CNAN$ PAM. Nucleotide T(− 7) and residues Thr1025 and Ala1027 are depicted by space-filling models. Hydrogen bonding and electrostatic interactions are shown as green dashed lines. **e** In vitro DNA cleavage activities of the wild-type BlCas9 (WT) and PAM recognition mutants. The linearized plasmid targets with the $T_3CCCAA$ PAM were incubated with the BlCas9–sgRNA complex at 37 °C for 2 min. Data are mean ± s.d. ($n = 3$). **f** In vitro DNA cleavage activities of WT BlCas9, the T1025A variant, and enBlCas9 toward DNA targets with different PAMs. The linearized plasmid targets were incubated at 37 °C for 2 min. Data are mean ± s.d. ($n = 3$).

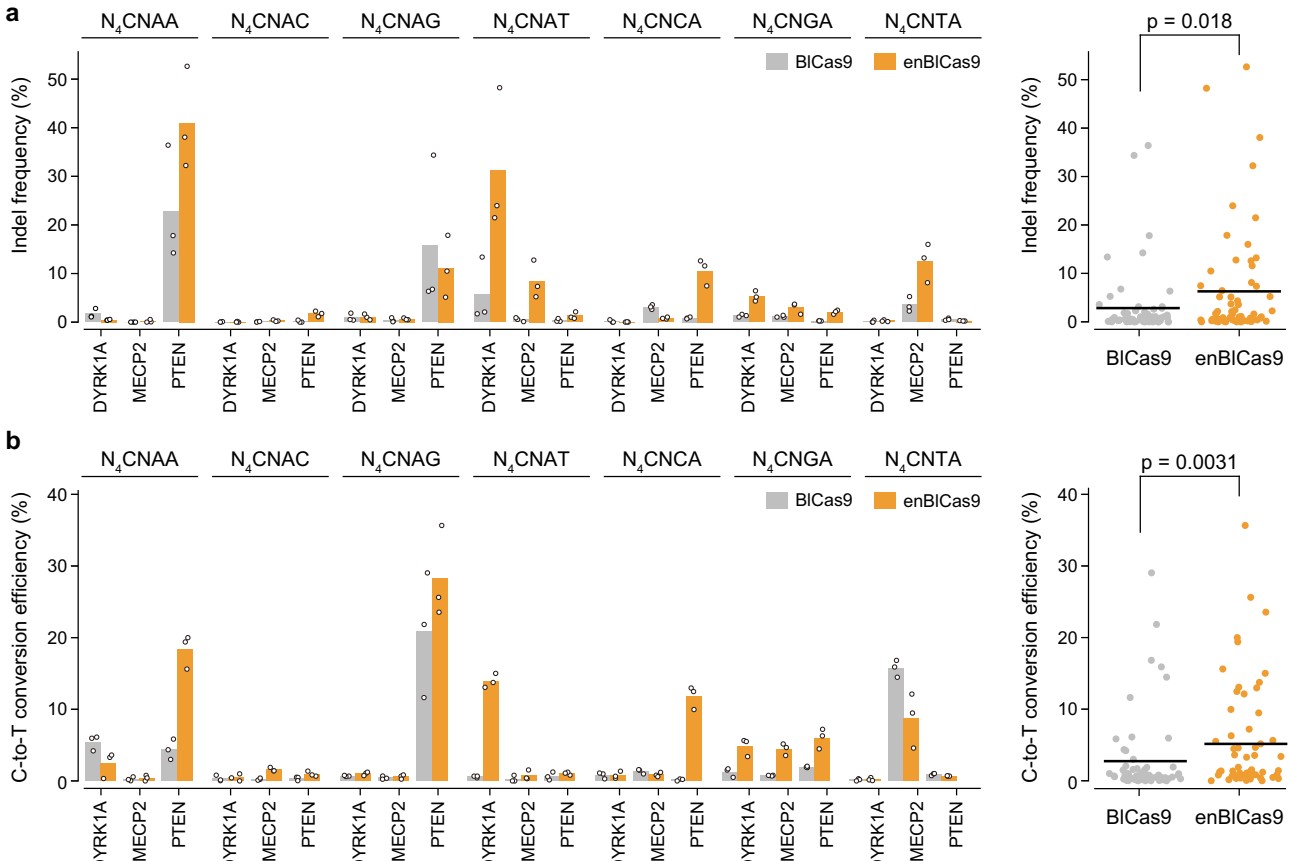

**Fig. 5 | Genome- and base-editing by BlCas9 and enBlCas9. a, b** Efficiencies of indel formation (**a**) and C-to-T conversion (**b**) by BlCas9 (WT) (gray) and enBlCas9 (orange) at endogenous target sites in HEK293Ta cells (*n* = 3). The *p*-value was calculated by the Mann–Whitney *U*-test.

diluted to 300 mM with 50 mM Tris-HCl, pH 8.0. The protein was mixed with 0.1 ml SP Sepharose High Performance resin (GE Healthcare) equilibrated with buffer C (20 mM Tris-HCl, pH 8.0, and 300 mM NaCl), and the mixture was loaded into a Poly-Prep Column (Bio-Rad). The protein was eluted with buffer D (20 mM Tris-HCl, pH 8.0, 1 M NaCl, and 1 mM DTT). The purified proteins were stored at –80 °C until use. The 110–113-nt sgRNAs (containing 20–23 nt guides) were transcribed in vitro with T7 RNA polymerase, and purified by 10% denaturing (7 M urea) polyacrylamide gel electrophoresis (Supplementary Table 1). RNA bands were excised from the gel and recovered with an Elutrap System (GE Healthcare). The sgRNAs were loaded onto a PD-10 desalting column (GE Healthcare), eluted with buffer E (10 mM Tris-HCl, pH 8.0, and 150 mM NaCl), and then stored at –20 °C until use.

For crystallization, we prepared the BlCas9-ΔHNH variant lacking the HNH domain (residues 505–670), in which Thr504 (RuvC-II) and Tyr671 (RuvC-III) are connected by a GGGSGG linker. The N-terminally His₆-tagged BlCas9-ΔHNH was expressed in *E. coli* Rosetta 2 (DE3) and prepared as described above. The *E. coli* cells were resuspended in buffer F (50 mM Tris-HCl, pH 8.0, 20 mM imidazole, and 300 mM NaCl), lysed by sonication, and then centrifuged at 40,000 g for 30 min. The supernatant was mixed with 4 ml Ni-NTA Superflow resin equilibrated with buffer F, and the mixture was loaded into an Econo-Column (Bio-Rad). The protein was eluted with buffer G (50 mM Tris-HCl, pH 8.0, 300 mM imidazole, and 300 mM NaCl). The eluted protein was loaded onto a HiTrap SP HP column (GE Healthcare) equilibrated with buffer C. The protein was eluted with a linear gradient of 0.3–2 M NaCl. To remove the His₆-SUMO-tag, the protein was mixed with TEV protease, and then dialyzed at 4 °C overnight against buffer H (20 mM Tris-HCl, pH 8.0, 40 mM imidazole, and 500 mM NaCl). The protein was passed through the Ni-NTA column equilibrated with buffer H. The protein was further purified by chromatography on a

HiLoad 16/600 Superdex 200 column (GE Healthcare) equilibrated with buffer I (10 mM Tris-HCl, pH 8.0, 500 mM NaCl, and 1 mM DTT).

The 110 nt sgRNA was transcribed in vitro with T7 RNA polymerase, using a partially double-stranded DNA template (Supplementary Table 2). The transcribed RNA was purified by 8% denaturing (7 M urea) polyacrylamide gel electrophoresis. The target and non-target DNA strands were purchased from Sigma-Aldrich (Supplementary Table 2).

**Crystallography**

The BlCas9-ΔHNH–sgRNA–target DNA complex was reconstituted by mixing the purified BlCas9-ΔHNH protein, the 110 nt sgRNA, the 28 nt target DNA strand, and the 8 nt non-target DNA strand (the T₃CCAAA PAM) (molar ratio, 1:1.5:2.3:2.5). The BlCas9-ΔHNH–sgRNA–DNA complex was purified by gel filtration chromatography on a Superdex 200 Increase column (GE Healthcare) equilibrated with buffer J (10 mM Tris-HCl, pH 8.0, 150 mM NaCl, and 1 mM DTT). The purified BlCas9-ΔHNH–sgRNA–target DNA complex was crystallized at 20 °C, using the hanging-drop vapor diffusion method. Crystals were obtained by mixing 1 μl of complex solution (A₂₆₀ ₙₘ = 25) and 1 μl of reservoir solution (200 mM sodium-acetate (pH 4.5), 15%–20% PEG 500 MME, 200 mM ammonium sulfate, and 10 mM strontium chloride). X-ray diffraction data were collected at 100 K on beamlines BL41XU at SPring-8 and X06SA at SLS. The crystals were cryoprotected in reservoir solution supplemented with 20% ethylene glycol. X-ray diffraction data were processed using DIALS[26]. Finally, 11 datasets were merged using KAMO[27] and XSCALE[28]. The structure was determined by molecular replacement with Molrep[29], using the coordinates of CjCas9 (PDB:5X2D)[11] as the search model. The model was rebuilt using Buccaneer[30], followed by interactive model rebuilding using COOT[31] and structural refinement using phenix.refine[32] and Refmac5[33,34]. An AlphaFold2-predicted model using ColabFold[35,36]

facilitated the model building of less-ordered regions. Data collection statistics are summarized in Table 1. Structural figures were prepared using CueMol (http://www.cuemol.org).

## In vitro cleavage assay

The EcoRI-linearized pUC119 plasmid (100 ng, 4.7 nM), containing the 23 nt target sequence and the PAMs (Supplementary Table 1), was incubated at 37 °C for 0.5–5 min with the BlCas9–sgRNA complex (100 nM) in 10 μl of reaction buffer, containing 20 mM HEPES, pH 7.5, 100 mM KCl, 2 mM $MgCl_2$, 1 mM DTT, and 5% glycerol. The reactions were stopped by the addition of quench buffer, containing EDTA (20 mM final concentration) and Proteinase K (40 ng). The reaction products were resolved, visualized, and quantified with a MultiNA microchip electrophoresis device (SHIMADZU).

## PAM identification assay

The PAM identification assay was performed as described previously[21]. The PAM library (100 ng), containing eight randomized nucleotides downstream of a 22 nt target sequence (Target 1), was incubated at 37 °C with the purified BlCas9 (WT and enBlCas9) (100 nM) and the sgRNA22 in 10 μl of reaction buffer, containing 20 mM HEPES, pH 7.5, 100 mM KCl, 2 mM $MgCl_2$, 1 mM DTT, and 5% glycerol. The reactions were stopped by the addition of quench buffer, containing EDTA (20 mM final concentration) and Proteinase K, and then purified using a Wizard DNA Clean-Up System (Promega). The purified DNA samples were amplified for 25 cycles, using primers containing common adapter sequences. After column purification, each PCR product (~5 ng) was subjected to a second round of PCR for 15 cycles, to add custom Illumina TruSeq adapters and sample indices. The sequencing libraries were quantified by qPCR (KAPA Biosystems), and then subjected to paired-end sequencing on a MiSeq sequencer (Illumina) with 20% PhiX spike-in (Illumina). The sequencing reads were demultiplexed by primer sequences and sample indices, using NCBI Blast + (version 2.8.1) with the blastn-short option. For each sequencing sample, the number of reads for every possible 8 nt PAM sequence pattern (48 = 65,536 patterns in total) was counted and normalized by the total number of reads in each sample. For a given PAM sequence, the enrichment score was calculated as log2-fold enrichment as compared to the untreated sample. PAM sequences with enrichment scores of −2.0 or less were used to generate the sequence logo representation, using WebLogo (version 3.7.1)[37]. The cumulative distribution and histogram of the read count of each PAM in the unedited sample confirmed that the plasmid library has sufficient coverage for the individual PAM sequences.

## Genome- and base-editing analyses in human cells

Genome- and base-editing analyses were performed in triplicate, according to the protocol described previously[38]. Briefly, HEK293Ta cells were maintained in DMEM (Sigma) supplemented with 10% (v/v) fetal bovine serum (FBS) (Thermo Fisher Scientific) and 1% Penicillin-Streptomycin (Sigma), at 37 °C in a 0.05% $CO_2$ atmosphere. HEK239Ta cells were seeded at $5 \times 10^3$ cells per well in collagen I-coated 96-well plates, 24 h prior to transfection. HEK239Ta cells were transfected with a BlCas9 plasmid or a BlCas9-derived base-editor plasmid (120 ng) and an sgRNA plasmid (40 ng), using Polyethylenimine Max (Polysciences) (1 mg/ml, 0.5 μl) in PBS (50 μl) (Supplementary Table 3). The cells were harvested 3 days after transfection, treated with 50 mM NaOH (100 μl), incubated at 95 °C for 10 min, and then neutralized with 1 M Tris-HCl, pH 8.0 (10 μl). The obtained genomic DNA was subjected to two rounds of PCR, to prepare the library for high-throughput amplicon sequencing. Genomic regions targeted by sgRNAs were PCR-amplified to add custom primer-landing sequences (Supplementary Table 4). The PCR products were purified by AMPure XP magnetic beads (Agencourt), and then subjected to a second round of PCR to attach the custom Illumina TruSeq adapters with sample indices. After size-selection by agarose gel electrophoresis and column purification, the sequencing libraries were quantified using a KAPA Library Quantification Kit Illumina (KAPA Biosystems), multiplexed, and subjected to paired-end sequencing (600 cycles), using a MiSeq sequencer (Illumina) with 20% PhiX spike-in (Illumina). The sequencing reads were demultiplexed, based on sample indices and primer sequences. Using NCBI BLAST + (version 2.6.0) with the blastn-short option, the sequencing reads were mapped to the reference sequences to identify indels and substitutions in the target regions. To remove common PCR errors and somatic mutations, we deleted sequencing reads containing mutations (>1% frequency) commonly observed in the control samples from the edited samples, and then normalized the editing frequencies for the target sites by subtracting the mutation frequencies of the control samples from those of the edited samples.

## Statistics and reproducibility

In vitro cleavage experiments were performed at least three times. Data are shown as mean ± s.d. ($n = 3$). Kinetics data were fitted with a one-phase exponential association curve, using Prism (GraphPad).

## Data availability

The atomic coordinates of the BlCas9–sgRNA–target DNA complex have been deposited in the Protein Data Bank, with the accession number PDB: 8X5V. The source data behind the graphs in the paper can be found in Supplementary Data 1–4. Any remaining information can be obtained from the corresponding author upon reasonable request.

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

## Acknowledgements

We thank the beamline scientists at X06SA of the Swiss Light Source and BL41XU of SPring-8 for assistance with data collection. We also thank Dr. Takanori Nakane for assistance with the structure determination. H.N. is supported by JSPS KAKENHI Grant Numbers 21H05281 and 22H00403, the Takeda Medical Research Foundation, the Inamori Research Institute for Science, and JST, CREST Grant Number JPMJCR23B6. O.N. is supported by AMED Grant Numbers JP223fa627001 and JP19am0401005, the Platform Project for Supporting Drug Discovery and Life Science Research (Basis for Supporting Innovative Drug Discovery and Life Science Research (BINDS)) from AMED under Grant Numbers JP23ama121002 (support number 3272) and JP23ama121012, and the Cabinet Office, Government of Japan, Public/Private R&D Investment Strategic Expansion Program (PRISM), Grant Number JPJ008000, and Cross-ministerial Strategic Innovation Promotion Program (SIP), "Technologies for Smart Bio-industry and Agriculture" (funding agency: Bio-oriented Technology Research Advancement Institution).

## Author contributions

T.N. performed biochemical experiments and crystallized the complexes, with assistance from R.N., S.O., Y.S. and H.N.; T.N., K.Y. and H.N. determined the crystal structures; S.I., H.M. and N.Y. performed cell biological experiments; R.N. and H.N. wrote the manuscript with assistance from S.I. and O.N.; H.N. and O.N. supervised all of the research.

## Competing interests

O.N. is a co-founder, board member, and scientific advisor of Curreio. The remaining authors declare no competing interests.
