## [Peer Review File · Communications Biology]

Reviewers' comments:

Reviewer #1 (Remarks to the Author):

In the manuscript, the authors analyzed the crystal structure of the BICas9–sgRNA–target DNA complex, offering high-resolution insights into the sgRNA architecture and PAM recognition. The authors further engineered BICas9 to expand the targeting scope. Although several Cas9 crystal structures have been analyzed, this work is still useful for understanding the diversity of natural Cas9 nucleases. Two papers related to BICas9 have been published. However, the current manuscript ignored an important reference paper published by Gao et al. (PMID:33195228). In that paper, Gao et al. already optimized the guide RNA lengths, and also tested BICas9 activity in mammalian cells. Therefore, the manuscript should be rewritten based on that paper. To facilitate communications, the Cas9 name should be kept as its original name (BlatCas9). Several additional questions have been identified:

1. In Fig1a, the 23-nt guide is more efficient than the 22-nt guide. In Fig1b, however, the 22-nt guide is more efficient than the 23-nt guide during 0-4 minutes. Please discuss it.
2. In the experiment testing the optimal sgRNA length, the authors conducted the experiment in vitro and focused on a single locus. To enhance the robustness of the findings, the authors should test different sgRNA lengths at multiple endogenous loci.
3. The site E904 of BICas9 is not presented in the full article or figures. Despite the author's statement that the E904R/T1025A variant increases the activity of BICas9, an explanation for this result is lacking. We suggest the authors provide more details and clarify the role of the E904 site in the manuscript or figures.
4. Regarding the single mutation of E904R, does it enhance cleavage activity or expand the target range compared to WT BICas9? Additional insights into the functional consequences of this mutation would be valuable for a comprehensive understanding.

Reviewer #2 (Remarks to the Author):

Overall, nice work presenting a structure of a novel Cas9 protein bound to multiple RNA molecules. I think that this data would be of interest to a broad audience. The structural data appear well done and clear and the PAM site recognition data are well thought out although a little confusing to read. This reviewer believes that the authors over state some of the gene editing significance without fully backing up these claims. This, however, is a minor critique and can be rectified without further experiments.

Lines 118-124 While I think that what is written is supported by the data, it is confusing as written in this reviewer's opinion, consider revising.

Lines 232 -236 Consider softening these statements. The engineered protein was moderately better in one cell culture line. This is not a particularly strong change in efficiency and cannot be applied broadly to mammals. In addition, why these three genes were chosen is unclear? It should also be mentioned that only of the PAM sites tested were mutable in all three genes tested, maybe two.

Line 278 should be “ a more compact editing toolbox” Compact is a subjective term.

In the discussion, in this reviewer’s opinion, more time space should be dedicated to why this new CRISPR holds an advantage over other CRISPR systems. One would think that the altered PAM site specificity would be a major one but that is briefly touched on. For instance, how many genes and how many times per gene can this system target the human genome. Are there previously untargetable regions that can now be targeted. A broader analysis or discussion of this would increase the significance of the findings.

Figure 5 How the significance was calculated is not stated in the figure legend – should be noted that the vast majority of PAM sites are targeted at very low levels or not at all. This makes the P values less relevant in terms of biological significance.

Reviewer #1:

In the manuscript, the authors analyzed the crystal structure of the BICas9–sgRNA–target DNA complex, offering high-resolution insights into the sgRNA architecture and PAM recognition. The authors further engineered BICas9 to expand the targeting scope. Although several Cas9 crystal structures have been analyzed, this work is still useful for understanding the diversity of natural Cas9 nucleases. Two papers related to BICas9 have been published. However, the current manuscript ignored an important reference paper published by Gao *et al.* (PMID:33195228). In that paper, Gao *et al.* already optimized the guide RNA lengths, and also tested BICas9 activity in mammalian cells. Therefore, the manuscript should be rewritten based on that paper. To facilitate communications, the Cas9 name should be kept as its original name (BlatCas9). Several additional questions have been identified:

We thank the reviewer for the comments. We have cited Gao *et al.* (PMID:33195228) in the revised manuscript. We also have included the original name "BlatCas9" on its first mention, but we would like to use the term "BICas9" throughout the manuscript, as "SpCas9" and "SaCas9" are widely used.

Comments:

1. In Fig1a, the 23-nt guide is more efficient than the 22-nt guide. In Fig1b, however, the 22-nt guide is more efficient than the 23-nt guide during 0-4 minutes. Please discuss it.

Thank you for the comment. We noticed that the order of the gel in Fig. 1a was incorrect. We have carefully included the correct figure in the revised manuscript (Extended Data Fig. 1a).

2. In the experiment testing the optimal sgRNA length, the authors conducted the experiment *in vitro* and focused on a single locus. To enhance the robustness of the findings, the authors should test different sgRNA lengths at multiple endogenous loci.

Thank you for your valuable comment. We designed sgRNAs targeting two additional, different 20–23-nt sequences and performed *in vitro* cleavage experiments. BICas9 exhibited varying *in*

in vitro cleavage activities toward the different targets, consistent with the indel and C-to-T conversion efficiencies in human cells (Fig. L1). Nonetheless, the sgRNA22 was optimal toward all three target sequences (Fig. L1). We have added these results in the revised manuscript (Fig. 1a).

Fig. L1 | Optimal guide length for BICas9.

In vitro DNA cleavage activities of BICas9 with the 20–23-nt guide sgRNAs toward three different sequences. The linearized plasmid target bearing the T₃CCGAA PAM was incubated with the BICas9–sgRNA complex at 37°C for 0.5, 1, 2, and 5 min. The cleavage products were then analyzed by a MultiNA microchip electrophoresis system. Data are mean ± s.d. ($n = 3$).

3. The site E904 of BICas9 is not presented in the full article or figures. Despite the author's statement that the E904R/T1025A variant increases the activity of BICas9, an explanation for this result is lacking. We suggest the authors provide more details and clarify the role of the E904 site in the manuscript or figures.

Thank you for the comments. Glu904 in the WED domain is located close to the PAM duplex, and modeling suggested that Arg904 forms a new interaction with the phosphate backbone of dA(-1), thereby enhancing the DNA cleavage activity (Fig. L2). We have included this figure in Extended Data Fig. 6a.

Fig. L2| Location of the E904R mutation.

Mapping of E904R onto the BICas9–sgRNA–target DNA complex. Glu904 in the WED domain is close to the PAM duplex. Modeling suggested that Arg904 interacts with the phosphate backbone of dA(-1).

4. Regarding the single mutation of E904R, does it enhance cleavage activity or expand the target range compared to WT BICas9? Additional insights into the functional consequences of this mutation would be valuable for a comprehensive understanding.

According to your insightful comment, we purified the E904R mutant, and compared its *in vitro* cleavage activities to those of wild-type BICas9 toward seven target DNAs with different PAMs. The E904R mutant efficiently cleaved all targets, except for the optimal T₃CCCAA PAM, compared to the wild-type BICas9 (Fig. L3). This result suggests that the E904R mutation enhances the cleavage activity of BICas9, thereby reducing its PAM constraint. We have added this result in the revised manuscript (Extended Data Fig. 6b).

Fig. L3| *In vitro* cleavage activities of the E904R mutant.

In vitro DNA cleavage activities of BICas9 (WT) and the E904R mutant toward DNA targets with different PAMs. The linearized plasmid targets were incubated with the BICas9–sgRNA complex at 37°C for 2 min, and the cleavage products were then analyzed by a MultiNA microchip electrophoresis system. Data are mean \pm s.d. ($n = 3$).

Reviewer #2:

Overall, nice work presenting a structure of a novel Cas9 protein bound to multiple RNA molecules. I think that this data would be of interest to a broad audience. The structural data appear well done and clear and the PAM site recognition data are well thought out although a little confusing to read. This reviewer believes that the authors over state some of the gene editing significance without fully backing up these claims. This, however, is a minor critique and can be rectified without further experiments.

We thank the reviewer for the positive comments. Our responses to the comments raised by the reviewer are as follows.

Comments:

1. Lines 118-124 While I think that what is written is supported by the data, it is confusing as written in this reviewer's opinion, consider revising.

Thank you for the comment. BICas9 recognizes a long, promiscuous PAM, and thus the sentence was confusing. To clarify this, we have underlined the PAM positions of interest in each sentence, such as T₃CCCAA. In addition, we have changed the conclusion statement from “Together, these results indicated that BICas9 recognizes N₄CNDN rather than N₄CNDD as the PAM, and requires A at either the 7th or 8th position.” to “Together, these results indicated that BICas9 recognizes N₄CNDN as the PAM, and prefers A at both the 7th and 8th positions.” in the revised manuscript.

2. Lines 232 -236 Consider softening these statements. The engineered protein was moderately better in one cell culture line. This is not a particularly strong change in efficiency and cannot be applied broadly to mammals. In addition, why these three genes were chosen is unclear? It should also be mentioned that only of the PAM sites tested were mutable in all three genes tested, maybe two.

Thank you for your insightful comment. To validate the genome editing activities of BICas9 and enBICas9, we selected three genes, DYRK1A, MECP2, and PTEN, which are commonly used in genome editing experiments with CRISPR-Cas enzymes. Our engineered enBICas9 exhibited higher indel frequencies than BICas9 at several target sites. However, as you pointed out, these results are based on a single cell line, and the observed improvement is moderate. Consequently, we have moderated our statements regarding the utility of enBICas9.

3. Line 278 should be “a more compact editing toolbox” Compact is a subjective term.

According to your comment, we have modified the text in the revised manuscript.

4. In the discussion, in this reviewer's opinion, more time space should be dedicated to why this new CRISPR holds an advantage over other CRISPR systems. One would think that the altered PAM site specificity would be a major one but that is briefly touched on. For instance, how many genes and how many times per gene can this system target the human genome. Are there previously untargetable regions that can now be targeted. A broader analysis or discussion of this would increase the significance of the findings.

Thank you for the valuable comment. While several Cas9 and Cas12 orthologs have been reported to exhibit genome-editing activities in mammalian cells, most Cas9 and Cas12 orthologs require G- and T-rich sequences as their PAMs, respectively, thereby restricting their targetable genomic loci. In contrast, enBICas9 can induce genome- and base-editing at target sites that lack G or T, potentially enabling applications in the treatment of genetic diseases that were previously inaccessible. In addition, since enBICas9 (1,092 residues) is much smaller than SpCas9 (1,368 residues), it may be feasible to package enBICas9 or enBICas9-AID into a single AAV vector, facilitating *in vivo* therapeutic genome- and base-editing. According to your comment, we have included these statements in the Discussion section of the revised manuscript.

5. Figure 5 How the significance was calculated is not stated in the figure legend – should be noted that the vast majority of PAM sites are targeted at very low levels or not at all. This makes the P values less relevant in terms of biological significance.

Thank you for the comments. The p-values in Fig. 5 were calculated by the Mann–Whitney U test. According to your comments, we have added this information to the legend in Fig. 5.

REVIEWERS' COMMENTS:

Reviewer #1 (Remarks to the Author):

The author has addressed all of my concerns, and the revised manuscript has improved significantly. There is only one that needs improvement. Please indicate in the diagram what genes target 1-3 refers to.